# Where is communication breaking down? Narrative tensions in obesity-in-pregnancy clinical encounters

Rachel Dadouch[1,2,3]*, Sarenna Lalani[2,4], Rory Windrim[2], Cynthia Maxwell[2,5], John Kingdom[2], Rohan D'Souza[1,2,3,6‡], Janet Parsons[1,7,8‡]

1 Institute of Medical Sciences, Temerty Faculty of Medicine, University of Toronto, Toronto, Ontario, Canada, 2 Division of Maternal-Fetal Medicine, Department of Obstetrics & Gynaecology, Mount Sinai Hospital, University of Toronto, Toronto, Ontario, Canada, 3 Department of Obstetrics & Gynaecology, McMaster University, Hamilton, Ontario, Canada, 4 Department of Medicine, School of Medicine, Queen's University, Kingston, Ontario, Canada, 5 Women's College Hospital, Toronto, Ontario, Canada, 6 Department of Health Research Methods, Evidence and Impact, McMaster University, Hamilton, Ontario, Canada, 7 Department of Occupational Science & Occupational Therapy, Temerty Faculty of Medicine, University of Toronto, Toronto, Ontario, Canada, 8 Li Ka Shing Knowledge Institute, St. Michael's Hospital, Unity Health Toronto, Toronto, Ontario, Canada

‡ RD and JP are joint senior authors on this work.
* dadouchr@mcmaster.ca

**Data Availability Statement:** All relevant data are within the manuscript.

## Abstract

There are numerous biomedical and psychosocial challenges associated with obesity in pregnancy that impede communication between healthcare providers (HCPs) and patients. We conducted a narrative study informed by stigma theory to understand specific areas of communication breakdown in obesity-in-pregnancy clinical encounters. Sixteen patients and 19 HCPs participated in in-depth, semi-structured interviews. We explored how participants positioned obesity-in-pregnancy clinical encounters within their broader narratives. Employing narrative analysis, we identified five narrative tensions contributing to communication challenges: 1) obesity as *a detriment to health* versus *an acceptable biologic variation*; 2) *obesity as the result of personal choice* versus *the result of uncontrollable circumstances*; 3) *a regular pregnancy* versus *a high-risk diagnosis;* 4) *a typical and problem-free clinical encounter* versus *a tremendously difficult clinical encounter*; and 5) *talking openly about Body Mass Index (BMI) and related co-morbidities* versus *sidestepping the topic*. How participants positioned themselves relative to prevailing societal discourses regarding obesity in general influenced these tensions. These narrative tensions revealed specific areas where communication is vulnerable to breaking down during the obesity-in-pregnancy clinical encounter. Participants' (both HCPs and patients) past experiences of clinical encounters–and the meanings they ascribe to them–shape subsequent encounters, and our analysis illuminates the complexities of this interactive space. This research has implications for improving clinical practice and education.

**Funding:** Frederick Banting and Charles Best Canada Graduate Scholarships – Doctoral Research Award (CGS-D) to Honour Nelson Mandela Canadian Institutes of Health Research (CIHR) Amount: $105,000 ($35,000 per year for 36 months) 2020 - 2023 Ontario Graduate Scholarship Program (OGS) University of Toronto, ON Amount: $15,000 ($5,000 per semester) 2019 - 2020 Endowment Award Obstetrics and Gynaecology University of Toronto Obstetrics & Gynaecology, Amount: $15,000 2019 - 2020 The funders had no role in study design, data collection and analysis, decision to publish, or preparation of the manuscript.

## Introduction

Obesity-in-pregnancy clinical encounters may be exceptionally difficult and uncomfortable for both healthcare providers (HCPs) and patients to navigate. Maintaining an optimal provider-patient relationship is a key priority, while minimal resources currently exist to help providers avoid unintended harm. Addressing this gap in care is an important priority since the prevalence of obesity is widespread across middle and high-income countries [1]. The World Health Organization reports that globally, 15% of women live with obesity [2], which has traditionally been defined by a body mass index (BMI) above $30kg/m^2$, though holistic classification systems are emerging that extend beyond the BMI criterion alone [3]. In Canada, 43% of women from 20 to 34 years old live with a BMI above $25kg/m^2$ [4].

Irrespective of pregnancy outcome, obesity alone is associated with a range of potential health care complexities including subfertility [5]. Obesity is now seen as a specific disease [6], or at least as a risk factor for life-threatening complications [2], at the level of governmental bodies [7] and healthcare institutions [8]. Accordingly, there is widespread weight loss recommendations as a means of mitigating against such adverse health outcomes. By contrast, the 'weight-neutral' approach does not prioritize weight loss per se, though focuses on body acceptance, individualized nutritional needs and physical activities that are enjoyable and sustainable [9].

In pregnancy, there are added challenges related to obesity. 4.14% of people with a BMI above 40 that become pregnant have pre-existing diabetes, while 8.86% also have chronic hypertension (versus 0.71% and 0.69%, respectively, of those with a BMI 18.5–24.9) [10]. Individuals with chronic hypertension have a substantially elevated risk of hypertensive disorders in pregnancy (e.g. preeclampsia), preterm birth, macrosomia, stillbirth [11], and postpartum haemorrhage, all of which increase the relative risk of maternal death [12]. Independent of pre-pregnancy health, several adverse health outcomes are increased, including: venous thromboembolism [13], wound infection [14], protracted labour [15], gestational diabetes, hypertensive disorders of pregnancy, caesarean deliveries (CDs) [10] and anaesthetic complications [16]; and for infants, congenital anomalies [17], premature birth, stillbirth, macrosomia, birth trauma, respiratory distress, lower Apgar scores, shoulder dystocia, hypoglycaemia, infection, neonatal intensive care unit admission and neonatal death [10]. Obesity-in-pregnancy care can be procedurally difficult, such as with the performance of ultrasound examinations, which can delay or prevent effective ultrasound-based prenatal diagnosis [18], and make fetal monitoring in labour challenging, resulting in the late diagnosis of and delayed interventions for fetal distress [19]. Furthermore, clinical decision making around mode of delivery for this patient group has unique considerations [20] that may require specialized anaesthetic and surgical expertise [21] and often cannot be immediately implemented. In the postpartum period, failed breastfeeding initiation is more common [22], as are depression and anxiety [23, 24]. As consequence of these aforementioned concerns, HCPs treat obesity in pregnancy as a condition that requires active management, which may be perceived as stigmatizing by patients, as described in qualitative studies [25]. Patients may feel that their BMI is seen as their main identity [26] commonly reporting insensitive comments and a sense of over-medicalization of their obesity [27]. They are aware of the societal discourse about obesity that persons experiencing obesity are lazy and overeat, which worsens fears and anxieties [27].

These contrasting differences between providers and patients have important implications for optimization of clinical encounters. In previous studies, patients have reported highly negative experiences [28–30], such as leaving a physical exam in tears, receiving sneers when different equipment was needed, being called *really obese* [28] or told that they should never have been pregnant [29]. There are also neutral and/or positive encounters, in which patients say

they are treated like everyone else [31], feel listened to [29] and respected [32]. Predominantly however, patient perspectives depict conversations with their HCPs as judgmental, pessimistic, critical, confusing or not transparent [33–37], and there is variation on whether the mention of weight is received well. In terms of the lack of communication, patients noticed that clinicians seemed afraid to broach weight-related subjects. They reflect that staying silent on weight topics is problematic [31], as this can cause false reassurance [38] and missed information, such as not knowing why there was a referral to a specialist clinic [28]. HCPs have shared that they are weary of broaching the topic of weight, and may walk around it or dilute it with other matters [39, 40].

There is no one communication guide or strategy that exists for the obesity-in-pregnancy clinical encounter. The '5As of Healthy Pregnancy Weight Gain' [41] focuses on gestational weight gain, and was modelled after a tool geared towards the non-pregnant population with obesity. Moreover, our review of the quantitative literature showed that research in this area predominantly focuses on physiological outcomes, rather than patient-oriented considerations. Physiological and clinical outcomes comprise of nearly 75% of all outcomes we identified [42], whereas, our systematic review of qualitative studies on obesity in pregnancy determined that merely 31% of outcomes were physiological/clinical, and 21% were related to emotional functioning/well-being and 15% to delivery of care [43]. This highlights the need to capture the patient experience in pregnancy research. No study specifically explores the lived experience of individuals with obesity in pregnancy, distinct from specific aspects of healthcare [27, 31, 32, 38] and/or weight-related behaviours [25, 26, 44], or studies have limitations such as not being solely geared to obesity [26]. Studies on these clinical encounters thus far have not explored *both* patient and HCP narratives in tandem. Therefore, exploring the narratives of HCPs and patients and ascertaining where they align and misalign, can help identify points of communication breakdown, which can help establish tangible and usable communication strategies. The aim of this research was to understand the communication challenges within obesity-in-pregnancy clinical encounters from the perspectives of both parties.

## Methodology

We conducted in-depth qualitative interviews to unpack obesity-in-pregnancy clinical encounters, elucidating the narrative tensions in effect that may be compromising communication. This study asked the research question, *Where is communication breaking down in the obesity-in-pregnancy clinical encounter based on the stories told by the HCPs and patients who participate in this encounter*? This study was approved by the research ethics boards of the hospital (MSH REB 17-0316-E) and university (00038538).

### Informing theory

Our study design is informed by stigma theory [45–47], which refers to the process by which people are seen as having a discredited attribute that society rejects and that spoils one's identity [46]. This has a detrimental effect on stigmatized people, emotionally, mentally and physically [48, 49]. In the context of obesity and pregnancy, weight stigma denotes excess weight as deviating from the "norm" [49], by which society overestimates individual control [50]. People are considered to be at fault, and genetic and environmental vulnerability to dysfunctional human weight regulation [50] is overlooked, despite the evidence that there is a substantial genetic influence on appetite behaviours [51, 52]. Enacted stigma, the degree of experienced stereotyping, prejudice and discrimination [53]; anticipated stigma, the stigma that one may expect; and internalized stigma, the shame felt on account of stigma endured; all exacerbate the feeling of powerlessness and reduced self-efficacy, which can contribute to unhealthy

behaviours and health outcome disparities [53, 54]. Weight stigma in obstetric care has been
evident with increasing BMI, wherein pregnant people more negatively perceive the healthcare
they receive, and HCPs perceive this patient demographic to be taking less care of themselves
[55]. Inherent in studying clinical encounters, this research is framed within the social rela-
tional tradition of stigma theory. The constant presence of weight stigma leads to its manifesta-
tion in communication between people [56], echoing Goffman's work that people assume
certain roles during face-to-face interactions based on enacting what is considered normal
[46].

## Methodological stance

Narrative inquiry is the methodological stance of this study [57]. We studies the stories of
patients and HCPs to understand the meaning they assign to their experiences [58], which is
in line with an interpretivist approach [59, 60]. Narrative methodology focuses on the content
of the stories shared, but also form and tone [61]. Interviews are understood as co-constructed
between the participant and interviewer [62], and the resulting analysis of stories tells us about
how participants view themselves and others, revealing important aspects of identity, which
illuminates their approaches in clinical encounters [63].

## Study design and setting

We conducted narrative interviews [57, 58] to explore obesity-in-pregnancy clinical encoun-
ters at a tertiary care hospital in Toronto, Canada. Apart from several initial interviews with
HCPs conducted in person, the majority of interviews took place via telephone due to the
COVID-19 pandemic.

## Sampling and recruitment

Using a combined approach of purposive and convenience sampling, we recruited pregnant
and postpartum people with a BMI greater or equal to 30kg/m$^2$ who were patients in the spe-
cialized antenatal clinic at the study facility [64]. Although low-risk obstetricians, family doc-
tors and to a smaller extent midwives from across Ontario refer patients with a BMI>30 to
this clinic, most referred patients either have a BMI>40 or a BMI>30 in the presence of a
comorbidity. This referral clinic is part of an obesity-in-pregnancy program that has several
size-inclusive modifications, such as larger blood pressure cuffs and operating tables. We also
purposively sampled HCPs who are clinically involved at the different stages of obesity-in-
pregnancy care at this same centre, including physicians, nurses and other allied HCPs.
Recruitment of patients took place in-person between January to March 2020 and remotely,
between September 2020 to May 2021. Recruitment of HCPs took place between December
2019 and February 2021.

## Data collection

Between January 2020 and July 2021, we conducted open-ended individual interviews [65]
with 19 HCPs and 16 patients. Upon obtaining informed written or verbal consent (in-person
or via telephone, respectively), data collection and analysis occurred in an iterative fashion, in
that they were simultaneous and informed subsequent interviews [65]. Verbal consent was
obtained for the interviews that took place via telephone during the COVID-19 pandemic in
efforts to maintain a pandemic-friendly protocol. Participants consented over the phone,
which was documented by the interviewer in writing, prior to commencing the audio-record-
ing. This process was approved by the ethics board, including an amendment that specified the

COVID-19 friendly modifications to the process. Once interviews commenced, the process followed: the narrative phase and the conversation phase [66, 67]. Discussions evoked experiences and stories, and encouraged talking about anything that in some capacity related to obesity-in-pregnancy clinical encounters [68]. There were two interview guides; one for patients and HCPs each. Each set of questions served as a guide only, and participants were free to offer their accounts in any way that felt comfortable for them, including covering the topics in different orders. Both guides served to ensure that all key information of interest to the research team were covered during the interview. The interview guide focused on understanding what this clinical encounter is like for HCPs and patients, with a focus on communication and narratives. With HCPs, we delved into their roles when caring for this patient population and their experiences talking about weight. With patients, we asked about their preferences on being approached about weight-related matters, and when appropriate, their knowledge of weight-related considerations in pregnancy care.

## Data analysis

We analyzed data about participants' experiences of obesity-in-pregnancy clinical encounters, as well as any stories from their pasts that influence current expectations, decisions and behaviours. We attended to stigmatizing attitudes and experiences, and considered how participants positioned themselves in relation to others in their stories, the interviewer and themselves [69]. Qualitative research analysis occurs in conjunction with data collection in an iterative process. The analysis process [66, 67, 70] entailed reading through the transcripts multiple times, familiarizing ourselves with the data, getting a sense of tone and form of the stories told, and creating memos of initial impressions and insights. This was followed by manual coding of the transcripts to develop an initial coding framework and descriptive themes based on team discussions; we then condensed these into a final iteration of succinct themes [65, 67, 71]. This resulted in a document of colour-coded themes with quotes from transcripts and analytic notes described within. Keeping close to the data and looking across the data corpus, we reviewed these themes and exemplar quotes to identify further patterns in the dataset. The first author then used notes from these discussions, referred back to the analysis document with themes and notes, and paired commonalities and differences between themes and between participant stories, to form the identified narratives [72], and the resulting typology of narratives [70, 73]. Comparing within and across groups (patients and HCPs), we noted similarities and differences between data, grouping them into broader stories [66]; attending to the impact that participants' past experiences had on their current decisions and behaviours, which allowed for the developmental process of organizing these findings into narrative tensions in obesity-in-pregnancy clinical encounters. We defined a narrative tension as a point of competing or conflicting difference in positions on certain topics, that help illuminate participants' ways of approaching this encounter. Analytic rigour was enhanced by multiple members of the team having reviewed and coded the transcripts, and discussing the developing framework together.

Having the majority of interviews conducted via telephone, the text in the transcripts was richer as shared information rested solely on verbal communication [74]. We selected the layout of the results based on what can be useful to those affected by these stories [70]: the participants in this encounter. Considering the unknowns that HCPs have about patients, and vice versa, we unraveled the results in a way that could help them understand each other better. Although this work does not encompass every person and every story that one may find on this topic, these data still provide a guide of what a patient or HCP may be aware of and now look for [70]. Upon synthesizing participant narratives, we chose certain examples from the

original work to represent in this publication as ones that stood out as important to include [70, 72].

In narrative interviewing and analysis, reflexivity is an integral component of analytic rigour, as the social positioning of the first author (RD) who led both data collection and analysis, inevitably influenced the study findings. Reflexivity and recognition of the researchers' role is an accepted component of research with an interpretive ontology [75]. Reflexive notes were maintained throughout the study conduct. One note for example, is that the beginning of the data collection and analysis processes may have been influenced by the fact that RD is a researcher in an academic and clinical environment, and is not a plus-size person. This may have resulted in commencing interviews with a better understanding of the HCP perspective rather than the patient perspective in these clinical encounters. RD attended to this throughout the analysis and used this to challenge her own assumptions concerning the experiences of patient participants.

## Results

### Participants

We interviewed 16 patients and 19 HCPs. Participant demographics are displayed in Tables 1 and 2. Eleven patients and five HCPs identified as 'Canadian (three or more generations)'. Their self-reports included identification by Continent (Europe, Eurasia), Nationality (Greek), Region (Caribbean), Province (French Canadian) and/or racial/ethnic group (Aboriginal, Métis, Persian, Jewish, Black or White/Caucasian). We made the decision to omit BMI from the participant characteristics, because we wanted to present the dataset as a whole; insofar as patient and HCP stories about these clinical encounters, and the various tensions that arise and exist, that those affected can be cognizant of. We deemed it a contributor to weight bias and stigma for this study specifically, to differentiate between the classes of obesity. While we understand it is relevant and integral information for many studies, for this research, we considered it otherwise.

**Table 1. Demographic and medical information of patient participants.**

| Characteristic | Patients (n = 16) |
|---|---|
| Number of participants (n) | 16 |
| Age in Years (median, range) | 33.5 (24–38) |
| Occupation, n (%) | |
| Employed | 11 (68.75) |
| Unemployed/Homemaker | 4 (25) |
| Student | 1 (6.25) |
| Education, n (%) | |
| High School | 2 (12.5) |
| College (Incomplete/Currently Enrolled) | 2 (12.5) |
| College | 5 (31) |
| University (Undergraduate or Master's) | 7 (44) |
| Parity, n (%) | |
| First ongoing pregnancy | 9 (56) |
| Spontaneous or induced abortion prior to this pregnancy | 3 (19) |
| One or more previous full-term pregnancy | 4 (25) |
| Gestational Age in Weeks (median, range) | 29 (14–35) |

**Table 2. Demographic information and professional roles of HCP participants.**

| Characteristic | HCPs (n = 19) |
|---|---|
| Number of participants (n) | 19 |
| Age in Years (mean, range) | 44.1 (29–68) |
| Professional Roles | |
| Physicians | 8 |
| Nurses | 7 |
| Midwives | 2 |
| Other | 2 |

## Narrative tensions

We identified five narrative tensions (Table 3). The first three we characterized as competing narratives related to participants' perceived risk of obesity or obesity in pregnancy and the understanding of its causes, and the last two refer to whether this encounter is problematic and addressing or avoiding the weight topic. We do not suggest the 'correctness' of one story over another; rather narrative tensions in participants' accounts that showcase the differences in how these topics are approached, which in turn affect communication in the encounter.

The following are the five narrative tensions in detail, that incorporate findings from both patients and HCPs.

**Narrative tension #1: Obesity is only a detriment to health versus accepting obesity.** Some participants discussed obesity insofar as correcting it, centering weight loss, and that people with obesity are unhappy and/or at an imminent health risk, no matter the individual patient's health profile. Other participants overlooked obesity, in that they could engage in topics that did not center ridding or denouncing weight, as a high BMI in and of itself was not perceived to be the most severe matter at hand. Participants who narratively fell in the former did so by characterizing obesity as either unwanted or unsafe. There were HCPs who shared statements such as: *"No one wants to be the way they are"* (ACBH4), or in reference to someone they know with obesity: *". . .I feel like you're killing yourself with a fork. . ."* (ACBH6). One patient shared the concern:

**Table 3. Five narrative tensions in obesity-in-pregnancy clinical encounters.**

| | Narrative Tensions | | |
|---|---|---|---|
| **Narrative Tension #1** | Obesity is Only a Detriment to Health:<br>• Focus on weight loss/gain<br>• Obesity unwanted/unsafe | versus | Accepting Obesity:<br>• Obesity and weight loss are not the be-all and end-all<br>• Obesity is not unhealthy |
| **Narrative Tension #2** | Obesity is the Result of Personal Choices | | Obesity Results from a Complex, Uncontrollable Interplay between Genetics and Environment |
| **Narrative Tension #3** | Obesity in Pregnancy is a Regular Pregnancy | | Obesity in Pregnancy is a High-Risk Diagnosis |
| **Narrative Tension #4** | The Obesity-in-Pregnancy Clinical Encounter is Similar to Any Other | | The Obesity-in-Pregnancy Clinical Encounter is Tremendously Difficult:<br>• Health risks<br>• Challenging healthcare delivery<br>• Uncertainties around how to act and what the other is thinking<br>• Communication challenges |
| **Narrative Tension #5** | Addressing the Weight Topic: Directly Discussing Having an Elevated BMI and Related Co-Morbidities | | Avoiding the Weight Topic<br>• Beating around the bush or substituting it altogether with indirect words/topics |

*"But what about ten years down the line, five years down the line? You don't know what's going to happen if you don't take care of the weight right now. . ."* (ACBP11)

The long-term effects of obesity are emphasized, which set the tone for the interview by presenting weight as a point of discussion solely in terms of gaining or losing pounds, just as one may when not pregnant. This also included during prompts about communication. For example, when asked about weight being a topic of conversation, one patient shares:

*"Well, they haven't really because I haven't really gained weight."* (ACBP5)

On the other end of this narrative spectrum, some participants (including both HCPs and patients) accepted obesity by not predominantly centring risks and the need to rid obesity. They talked about the differences in their care role or patient experience for a plus-size pregnancy, social experiences, health co-morbidities and/or reflecting on weight-related communication. One patient expressed their desire to participate in this study because they feel obesity in pregnancy rarely is explored beyond the scope of weight loss:

*". . .no one talked about plus size and health care. No one talked about it, unless it's like, 'you're fat, here's how to lose weight'. There's a lot of conversations about that, but there's not a lot of. . . support where it's like, 'okay, you're fat or you're bigger, here's how you can still do these things'[. . .] A lot of the time it's, 'you're this amount of weight[. . .]' and it's like, okay, but it's not always just numbers[. . .]"* (ACBP6)

Obesity and weight loss are not the be-all-end-all according to this participant. Correspondingly, in another interview, a HCP notes that in clinical encounters they try:

*"[. . .]not to make [weight] the whole thing. Like I don't want to stigmatize them for having an elevated BMI. So I just make it one of the points that we talk about, I don't make it the focus of everything we do."* (ACBH12)

This HCP echoes the value of not emphasizing weight in their clinical relationship. These examples represent two participants, though several others shared that moving beyond the number is integral in obesity care. On a more exaggerated end of *Accepting Obesity*, are HCPs who do not see obesity as something that would present an issue ever. For example, some HCP participants were taken aback when asked whether they experience any communication challenges with this group of patients, and distanced themselves from obesity-specific challenges.

**Narrative tension #2: Obesity is the result of personal choices versus uncontrollable circumstances.** HCP and patient participants' understandings of obesity and how it comes about, inevitably affected the communication in present-day obesity-in-pregnancy clinical encounters. The narratives range from *obesity is caused by an individual making poor choices*, to *obesity being caused by various factors*, *largely beyond the individual's control*. These different ways of reflecting on obesity seemed largely influenced by one's personal life. The following HCP shares that obesity results from individuals making poor lifestyle choices. They note that motivation is integral within a broader talking point about their own personal journey with weight:

*"I know if you put your mind to it you can do it, so I always tell myself I don't look at it as an impossible thing."* (ACBH4)

This reflects a prevailing societal discourse about obesity more generally that places responsibility on the individual to combat obesity. This narrative was consistent for many of the participants that had responded with weight loss anecdotes in the previous narrative tension. One of the patient participants gave a detailed explanation of their dietary habits and weight gain value, unprompted:

*"Doctors can only advise, you can only encourage, it is up to the person to make a decision like, okay, I want to lose this weight[. . .]"* (ACBP11)

The viewpoint conflicts with the perspective that obesity results from a complex, uncontrollable interplay between genetics and environment. One HCP reflects that their:

*". . .education was a long time ago and it was very much, weight is kind of simple. Calories in, calories out. If you eat too much, you gain weight. If you don't exercise enough you gain weight. And, of course over the last 40 years, we've all understood how nuanced it is, and how it's really not that simple at all and that you really diminish people if you reduce their struggles to that kind of equation. . ."* (ACBH8)

Similarly, patient participants commented on the harm of others blaming them, and that they would like HCP support to find explanations for how they gained weight:

*"[. . .]I need that professional help to get me over whatever demons I may be dealing with inside[. . .] to look at food differently, at people's pasts. I grew up in a family of [many] kids, and when food was set on the table you need to grab as much as you can. . . or else you're not eating. I think that's where all my eating habits started from, and it just, went out of control that way[. . .] So, it's looking at people's history and finding out how they got there."* (ACBP1)

This patient highlights unrecognized psychosocial trauma that they link to their obesity, and wanting to determine influencing factors. Another patient participant similarly shared that they would appreciate if their underlying issues could be assessed via tests that can ascertain the degree to which environmental factors and/or genetic factors are at play (ACBP7).

**Narrative tension #3: Obesity in pregnancy: A regular pregnancy or a high-risk diagnosis?.** Some participants thought about obesity in pregnancy as high risk, and some did not. For some, this was seen as simply a regular pregnancy akin to any other, just in a larger body. When asked the reason for their referral to the specialized clinic, patients in this way of thinking would not mention obesity, but merely a comorbidity (or multiple) as the reason for referral:

*"My O.B. recommended me to [hospital name] because I have sleep apnea[. . .] [weight] hasn't [affected my pregnancy]. I guess I haven't had any complications at all, whatsoever. I don't feel out of breath at all. I don't feel like I can't do certain things because of my weight. I still was able to do everything. I'm still able to work. I'm still able to move around. So I don't think weight has been a factor in my pregnancy at all, except for nowadays where I'm having more back pains and it's getting harder. But other than that, I've been good."* (ACBP7)

Some HCPs also characterized these pregnancies similarly. At times when probed about the connection of certain risks or accommodations to obesity, there were HCPs who distanced their responses from the uniqueness of obesity or its high-risk nature in pregnancy. For example, after discussing some differences in care for this patient population, such as the need for

using blood thinners to prevent blood clots post-operatively, a HCP followed with, *"...but, that's for their protection"*. They would qualify statements with, *"[...] which has got nothing to do with their weight, it'd be maybe they were there for a high-risk reason..."* (ACBH2). The participant seemed to minimize differences between caring for patients with obesity and others, conveying the broader message that obesity in pregnancy looks no different. The question arises whether this group of HCPs are performatively painting this pregnancy in this manner, to avoid the image or action of stigmatizing patients.

In tension with this position, is seeing this pregnancy as high risk, in a positive or negative light. At times this was evident with patients early on, when asked why they were in specialized care:

*"Oh yeah, oh yeah, I'm an open book, I don't mind. So basically I'm overweight... so that was a high-risk factor."* (ACBP12)

HCPs would share the high-risk status by elaborating on the obesity-specific supports, accommodations and obstetric risks. In the following example, a HCP notes that:

*"...there are patients[...] who come in and bring in those experiences from before with them and feeling like we're going to be critical of them or that we are seeing it as a character flaw as opposed to, um, like a medical condition. So I find for me it helps to look at it as a risk factor or condition rather than like a statement about the patient."* (ACBH13)

This HCP characterizes obesity in pregnancy as simply a health condition in efforts to not deem it a moral failure, as with the master societal narrative, acknowledging the experiences of blame that these patients are potentially coming in with.

**Narrative tension #4: The obesity-in-pregnancy clinical encounter is typical and problem-free versus tremendously difficult.** The clinical encounter between HCPs and patients with obesity in pregnancy is one that some thought of as any ordinary clinical encounter, or uniquely difficult. In the former, HCPs recounted that this encounter is unremarkable; the same as any other patient population:

*"Honestly, I'm very comfortable around them. Doesn't make a difference to me if they're high BMI[...] I do the same thing. I don't think I do anything different for them."* (ACBH4)

At the forefront of narrative inquiry is examining why participants depict their stories the way they do. Accordingly, this raises the question analytically of how these HCPs are portraying themselves, considering that despite their claims of no changes in treatment or care, there *are* differences in specialized obesity-in-pregnancy care. Patients that exemplified this narrative position described these clinical encounters with conversations that went well or were neutral (ACBP12). This characterization may have been possible due to comparing to other clinical encounters that were not as smooth. Some patients expressed this idea, that traumatic experiences with weight comments made certain medical clinical encounters less harsh:

*"Like when I was a kid, like I would be like, oh, 'what's for dinner Dad?'... he would be like, 'you know, you could stand to miss a meal or two'[...] so like having a small conversation with the doctor is kind of nothing."* (ACBP8)

Weight stigma had been described as a constant presence, both within clinical encounters and beyond, which affects the meaning-making process of current encounters. Another patient juxtaposed prior clinical encounters with their new ones in specialized care:

*"The bed sizes[. . .] mostly the problem is like chairs with arm rests, that cut into you. . . It's, it's very uncomfortable[. . .] So there's quite a few things that people who are bigger have to go through[. . .] With Dr. [name]'s, it's extra wide so you can sit and feel comfortable talking to your doctor. You're not focused on the arm rails getting into your side, you're just actually talking to her about whatever issues you may have."* (ACBP1)

The participant's account is one of relief at finally being in a care setting that feels more accommodating.

The other narrative stance was characterizing this clinical encounter as tremendously difficult. One of the first major reasons is the health risks, mostly cited by HCPs, that some participants described are directly linked to the patient's BMI:

*"So there are times when the impact of their BMI is causing us to either have to take a particular action or has implications that may be negative either to her own health or to the pregnancy, um, that is the only reason that they're having this issue"* (ACBH6)

These health risks were also reported to negatively impact clinical decision making, such as one HCP explaining the hesitation around the decision to proceed with a vaginal birth after a previous caesarean delivery (ACBH11). For patients, these health risks may be the trigger of enacted stigma under the guise of healthcare. One patient shared that when asking their family doctor about healthy nutrition and sugar intake for their newborn and young children, their doctor responded that it made no difference what is fed to the child because they will eventually see what the parent (patient) eats and gain weight too (ACBP1). There is an extensive history of patients being mistreated in the healthcare setting or beyond, that culminated in patients' and HCPs' descriptions of dehumanizing conditions that patients endure before their encounter together. These include comments on their diet upon setting a patient up with medical equipment at the emergency department (ACBP9 –Patient) and being weighed in a shipping/receiving area. Outside the healthcare setting, some patients have been rejected socially and experienced name-calling (ACBP10). These health-related challenges and stigmatizing experiences are compounded by the difficulty to communicate comfortably and productively. One HCP succinctly speculates, *"how to say the necessary things that you do have to say, but in a way that's sensitive and not offensive and objective, as well."* (ACBH6). Another HCP participant similarly notes:

Interviewer: *"Where do you see the line between what is important to convey to patients that their weight is playing a factor in[. . .] certain outcomes and when is weight not necessarily needed to be discussed?*

Participant: *That's a great question. I don't think any of us know the answer to that."* (ACBH8)

The ramifications of the communication struggle were felt by patients, as one recounts being asked over the phone why their BMI is higher than it was in a previous pregnancy:

*"So the fact that she said it like that, I was kind of not wanting to come back. But I knew that she's just the person in place, she's not [doctor's name], right."* (ACBP1)

This very question was perceived negatively enough for the patient to consider not returning for obstetric care. Others narrated emotional responses to communication too:

*"I think that that phone call ended with me hanging up and then crying and going home from work because I was so upset[. . .] She mentioned all of the normal risks, like pre-eclampsia, so on and so forth, but then she ended it with you have a higher chance of having a stillborn baby at the end of your pregnancy. And I'm like, okay, that could have waited until we were in person when we could elaborate on that a little bit more so. Not at the very end of the conversation and just kind of tossing it in the list of risks. So that I understand everyone has those risks, it just increases with being plus. It increases with being a bunch of other things, not only plus, but that's why I said it kind of took a bit of a darker, more intense direction rather than a light meet and greet."* (ACBP3)

This patient felt the need to leave work and described that assumptions were made because of their size. In addition to negative experiences, patients alluded to a lack of communication, such as with the following example about being prescribed blood thinners:

*"But I don't know if that was um, if everybody gets that or if that's because I'm four hundred plus pounds. . . And they're doing that as a precaution."* (ACBP1)

This quote demonstrates that this patient suspects that the prescription *might* have something to do with weight but that they remain uncertain, as this has not been discussed with them. This could be interpreted as placing an undue burden on patients to make this connection themselves. Some HCPs may experience these uncertainties about what weight-related conversations patients have had, as shared in the following story:

*"If you have a cord prolapse, you have to get the baby out within less than a minute, just to give a sense. But this patient [. . .] had to get general anaesthetic which is a longer process for mums with elevated BMI, it can be more challenging because of airway, and then we had to get through her abdominal wall to get the baby out, and this was not at all one minute, it was well beyond that, and when I talked to the patient beforehand, I said, 'this is an emergency and I will tell you now that, by the time we get to the baby, it's very possible that we may not, you know–that the baby may not be with us anymore,' [. . .] and I phrased it to her in a way that was, I felt, you know kind of sensitive and all that to her and the emergency situation. But I just felt at the same time that I don't know how much she truly understands what this means and how quickly this needs to happen [. . .]And I think we all kind of feel that way. We are all kind of on our toes. Yeah, it's very intense. And that situation, regardless of communication is very stressful and dangerous and I think the added layer of like not knowing within seconds you have to ascertain, what has this patient been informed of before this setting, and that's challenging to navigate on top of everything."* (ACBH15)

This HCP showcases the dilemma around communication during clinical encounters, particularly in emergency scenarios. In summary, this narrative tension captures the distance between describing this encounter as unremarkable or uneventful, and ridden with challenges related to health risks, stigma and communication.

**Narrative tension #5: Talking about BMI and related co-morbidities versus sidestepping the topic.** This final narrative tension captures the weight topic specifically in these encounters: addressing versus avoiding weight in conversation. Some HCPs explained that they initiate talking about weight, simply in the spirit of openness and honesty, by explaining the use of

BMI to assess risk (ACBH8) or letting the patient know that an outcome (e.g. epidural failure) is attributed to having a higher BMI so the patient is not left wondering (ACBH18). Some highlighted the importance of timing. They choose to mention BMI early on to avoid any surprises later (ACBH13). For some HCPs however, addressing weight felt more so out of necessity rather than a choice to be open and informative:

*"You're over a certain BMI, you take DVT prophylaxis, as low molecular weight heparin injections and you do that either for three or six weeks depending on your BMI. I have no choice but to have that conversation."* (ACBH6)

Narratively, these HCPs may be representing their role passively upon addressing weight, to remove the possible perception that they are stigmatizing these patients. In terms of addressing weight from patients' perspectives, doing so at the beginning of care as well as throughout pregnancy may be perceived negatively. A patient describes that raising this topic is like:

*". . .kicking a dead horse[. . .] so that's the only thing I kind of find irritating is that they keep bringing it up[. . .] every single time I go there, it's always, it's the first thing they want to talk about–It's not one of the other issues, it's always the weight first."* (ACBP12)

This patient is clear that they find it irritating to encounter weight in conversation so many times during care. Other patients shared positive stories:

*"When I talk to [hospital] on the phone, I felt so much better because she explains, how to do the ultrasound on the back for the epidural. And they do all of these things to make sure that things are just a little bit more in place[. . .] I don't live in a world where I don't know I'm overweight. I know I'm overweight. For me, I know there comes complications with that. I know it can change at any moment or any time. I totally get all that. And it does make it a little more complicated to do ultrasounds when you have a bigger belly[. . .] it's just the execution of the information or the explanation of all of it[. . .] when [it's] explains more you're like, 'okay, I get it.' I'm not as offended."* (ACBP10)

This participant wants to be informed, which speaks to the value of a patient feeling in the know about obesity-related information. There were patient participants who specifically said they preferred that weight is addressed in clinical encounters, positioning themselves distinctly separate from other people with obesity who potentially would be offended. They cite health and ensuring a healthy baby as the driving reasons to raise the weight topic.

In contrast to addressing weight, is the avoidance of it. For example, some HCPs do not feel they need to rehash it, at the risk of offending:

*"Probably a lot of people are avoiding it because no one wants to offend. And if it doesn't necessarily help the situation, it's not useful to bring up, necessarily. . ."* (ACBH19)

*"[. . .]she's not gonna know that the equipment that we use is slightly different. She's not gonna know they're going to use a mobius ring to help visualize the uterus[. . .] She doesn't know [that's] part of the care that she gets because of her high BMI. She doesn't know that the lift that we use is different[. . .] But, so, she doesn't need to know that right?"* (ACBH8)

Avoiding these obesity-specific differences aloud is a reasonable choice, however this points towards the clinical encounter being on the HCP's terms. This power imbalance was accentuated by some patients sharing that they had weight-related concerns, but did not want to

vocalize them, so that they do not intervene with HCPs' roles (ACBP10); or recounted the backlash they faced in healthcare settings in the past for communicating these needs (ACBP1). Weight then is not addressed in conversation.

To avoid mentioning weight, some HCPs may code their language by talking about co-morbidities only, leaving out *weight*, or use humour and other strategies. To each other, in front of patients, one HCP mentions:

> *"Instead of 'I'm bringing a BMI patient. . .', we say, 'I'm bringing a patient of Dr. [specialist]'s with the PICO dressing [special type of caesarean wound dressing used mostly in patients with an elevated BMI] and we need a bariatric room.' So that kind of gives us the sign that we need to have."* (ACBH4)

The coded language culminates in an "act" (referring to Goffman's literature) [76] that may continue between HCPs, as they are still in the same setting as patients. Another example is the use of larger sized gowns that have patterns specific to the designated size. This allows the HCP to quickly find the one they need, instead of struggling to find a larger gown aloud (ACBH1). Some HCPs characterize these changes as a tool to convey that this is a welcoming space. In this regard, the HCP acknowledges weight stigma and not wanting to contribute to this.

Patients also describe the avoidance of weight, such as with the following participant who describes being turned away by their local hospital due to resource limitations for people with high BMIs. The obstetrician conveyed this message to them in the following way:

> *"Um, and the one sentence that will live on forever in my head was that the anesthesiologist would not be happy to see my file cross their desk."* (ACBP15)

The patient shares that the HCP skips over the weight-related explanation and utilizes another HCP's potential reaction as their way to express this, rather than addressing the weight topic directly. Avoiding the weight topic also had direct ramifications on the delivery of care and potentially, health outcomes, as one patient shared that their local, referring centre circumvented conversation about the BMI-related scan limitations at their institution, which caused confusion and ambiguity. They recommended the patient attend the specialized centre to complete the scan, however the specialized centre explained to the patient that these scans are typically done at the local centre. The patient was unsure why they were told opposing pieces of information, and after more back-and-forth, the anatomy scan was delayed by eight weeks:

> *I felt like I was put into a box. And because I didn't meet those criteria, even though I'm a healthy person, like everything has come back healthy. I felt like I was put in a box and not told the entire truth[. . .] I was very happy with [specialized hospital] at my appointment. I accepted it for what it is, I do my regular appointments with my OB and they're very lovely and I guess I just kind of accepted it for what it is. I just wish throughout the process it would have been a little bit more upfront about what it really was[. . .] So I just wish I would have been like, 'okay, you know what, you're overweight[. . .] [specialized hospital] is more equipped' and just go. And I would have been like, 'okay cool', you know what I mean?"* (ACBP10)

This patient expresses a sense of betrayal from the healthcare space. Patients generally have experienced confusion in these instances, including with another example, in reference to the use of a weight chart/table:

*". . .I believe I've kind of find out that it's more so about my weight and BMI. [Community hospital] didn't feel comfortable for me to give birth there because I guess they have a chart of what they can and can't accept and so they can't accept me, so they sent me to a referral to [specialized hospital] for the high-risk clinic and that's where I met you guys[. . .] I just wish throughout the process it would have been a little bit more upfront about what it really was."* (ACBP10)

There is a sense of disappointment that the weight topic was not an open one, as they are a passive participant in these encounters.

## Discussion

In this study, we interviewed 16 patients and 19 HCPs at a tertiary centre in Toronto, to exploring in-depth patient-centered narratives related to obesity using an approach informed by stigma theory [46]. We identified five narrative tensions related to obesity-in-pregnancy clinical encounters. These narrative tensions were not necessarily a product of whether a participant was a patient or a HCP, and they positioned themselves in relation to one another as well as to family, friends, and others with obesity; they also positioned themselves within the broader societal obesity discourse/s. The narrative tensions capture positions that were defined by stories and their meaning from previous personal and healthcare experiences (and not merely the happenings in the encounter itself).

In prior qualitative studies, contradictions in perspectives have been noted. For example, in a study focused on intrapartum care [77], some HCPs expressed that the approach to maintain a sense of normalcy for patients with obesity may be via less intervention, though others thought that more intervention is preferable in order to foster a 'normal' outcome. Our narrative study however, utilizes these varying perspectives and uniquely allows for nuance in this complex clinical encounter. In some instances, certain strategies may be better than others.

These five narrative tensions are inherently inter-connected. Differing perspectives on the etiology of obesity (#2) and whether it is a detriment to health (#1), thereby constituting a high-risk pregnancy (#3), could result in challenges on deciding whether to address or avoid pertinent discussions (#5) and make for difficult conversations and clinical encounters (#4). These tensions display the specific topics and positions to look out for, as there are numerous ways that they can contribute to communication breakdown. Narratively being on opposing sides of any or all of these tensions, can result in a barrier between HCP and patient in pregnancy care. For example, seeing obesity solely as a detriment to health may result in obesity being spoken about in different ways (weight disapproval versus weight acceptance/neutrality). This depends on one's stance, stemming from the degree to which they believe obesity is dangerous to health from perceptions developed before even entering this encounter. Similarly, regarding the etiology of obesity, HCPs should be aware that attributing obesity to one's personal choices could be perceived as assigning blame and that patients are seen as having 'failed'. Concerning whether obesity constitutes a regular pregnancy or a high-risk diagnosis, touches upon Goffman's dramaturgical analogy [76], that there is an uncertainty of how to act when one does not know what the other knows. If the two actors are not thinking of obesity in pregnancy in the same way, there may be a mismatch in the extent of information provided by the HCP and the expectation of the patient. Priming conversations with questions about weight and health, that establish whether the patient leans towards one narrative position over the other, can help improve communication between the two people.

Holding differing views on whether the pregnancy is high-risk influences shared decision-making with regard to screening, preventative strategies and treatment plans. A discordance

may also happen between HCPs, leading to colleagues approaching a clinical encounter with the same patient differently. The tension between obesity-in-pregnancy clinical encounters as typical or challenging can lead to varying approaches to communication, of either treading weight topics carefully (at times, too carefully) or with less concern (to the extent of not being as mindful and sensitive). Even when both parties believe that this clinical encounter is difficult to navigate, there are still differing concerns and distinctive challenges that are personal to each individual. This is intertwined with addressing versus avoiding weight, which inevitably compromises communication, as there may be misaligned approaches in the degree to which weight is mentioned, leaving either or both of the parties' needs unmet and the potential for exacerbating stigma.

## Implications

Overall, the narrative tensions inform where communication is vulnerable to breaking down in obesity-in-pregnancy clinical encounters, and can be used as tools and signs for clinicians to look out for in clinical practice.

Various literature and recommendations in the obesity-in-pregnancy research area encourage HCPs to be aware of their own biases, and this work provides a specific way of doing so. Seeing *obesity as only a detriment to health* versus *accepting one's obesity* allows for reflection on which position they lean towards. Listening for where the patient positions themselves throughout the interaction is also important. For example, if patients respond to or bring up weight only in terms of weight loss (which is generally less relevant in pregnancy), this may suggest concern about obesity as solely a status that confers risk that must be eliminated. The HCP can provide the patient with reassurance using factual information and simply, consolation and compassion. The obesity-in-pregnancy clinical encounter might provide an opportunity to counter 'dominant narratives' [78] that a plus-size body is only equated with risk, though this does not preclude HCPs from offering evidence-based and thorough information about weight. The Canadian Adult Obesity Clinical Practice guidelines [3] emphasize that weight is not a behaviour and therefore should not be the target of behaviour change, and there is a call by obstetrics and gynaecology groups as well (e.g. ACOG) to not assign blame to patients [79]. Based on this work, there is a need to for the healthcare community to resist the societal discourse surrounding individual responsibility as the sole predictor of obesity, and be aware of the multitude of factors that contribute to excess weight.

Whether a patient or HCP considers obesity a high-risk or regular pregnancy could depend on experience, perspective, values or the desire to avoid stigma. Aligning with it being a typical pregnancy, may come down to relative and absolute risk [80]. For example, a relative risk increase of 1.34 for CD between two obesity classes, still translates to an absolute increase of just under 5% (from 13.92% to 18.85%), which to a patient implies that over 80% will still have a vaginal birth [10]. The meaning of these numbers can be personal and subjective, leading to patients leaning away from the high-risk status, compared to how HCPs experience these numbers. This may be indicative for patients of how they want to be spoken to regarding obesity-related differences in care, however HCPs should also not assume that all patients would *not* consider their pregnancy as high-risk. Doing so has resulted in silences in the clinical encounter, whereby the patient is craving more information. This can be mediated via gleaning in the earliest stages of the encounter how patients view their pregnancy, perhaps by explicitly asking their impressions of pregnancy thus far, as was done in our interviews. Based on the perception of this pregnancy as high risk, the encounter may then be considered difficult or simple. The goal may not be to create a more uniform perspective amongst these groups, as everyone will have different stories and interpretations, however, a simple awareness of a

patient's perception and navigating that during communication is a modest path to improvement. In that regard, although efforts to not stigmatize are important in improving care, not acknowledging the challenges in this encounter can lead to an impasse in this area. There is a need to normalize discussing the challenges in obesity-in-pregnancy clinical encounters and embark on solutions, in a way that HCPs will not fear being labelled as stigmatizing. In care teams, this could mean supporting HCP colleagues in a professional setting to discuss obesity challenges.

The relative positioning within the aforementioned tensions consequently may affect if one addresses or avoids the weight topic in these encounters. HCPs can reflect on which approach they lean towards, and in which situations they do so. This can follow an attempt to prevent abrasive differences in clinical encounters, after assessing the approach that the patient may be taking. HCPs are typically the leaders in communication in the HCP-patient relationship, and accordingly choices around avoidance could have implications for the patient related to stigma as well. In leading these conversations, there is an ethical consideration around the power-imbalance that exists between clinicians and patients, and consequently, HCPs must be cognizant that addressing versus avoiding the weight topic may be less so in the control of patients. Lastly, it is constructive to supplement weight-focused statements with other communication, such that weight is not a disproportionately predominating focus of clinical encounters, and the patient's needs can be met holistically.

This research was not designed to generate specific recommendations in terms of exactly how HCPs should address every situation where obesity plays a role in pregnancy. Rather it has illuminated important aspects of clinical encounters that have real consequences for care. Care is fundamentally relational, and this work serves to illuminate how to approach therapeutic relationships nimbly, tailoring it more closely to patients' needs and communication styles.

## Strengths and limitations

One of the strengths of this work is its use of narrative inquiry to explore obesity in pregnancy and this clinical encounter. Furthermore, it includes both HCPs and patients, in a way that brings their perspectives together to demonstrate how both parties struggle with each tension. Consequently, this work reveals hidden, personal realities of this encounter, in terms of what is not being said, what assumptions are made and the implications for care. It presents dilemmas for what constitutes informed care for this population. The methodological strengths include an uncharacteristically large sample size for a narrative study [81] that addressed various perspectives and care roles. The fact that we used predominantly telephone interviews may have helped reduce potential body-related judgments and anticipated stigma. However, not having this study conducted in person may have posed limitations as well. It was not possible to analyze non-verbal cues and body language that could have enhanced our analysis. From a demographic perspective, while our sample included a fairly diverse group of participants, we did not recruit non-English speaking people. In addition, while a wide range of HCPs were included, some health disciplines were not represented, such as social workers, physiotherapists and haematologists. On the note of transferability, the conduct of this study took place at a tertiary centre and in keeping with narrative inquiry does not involve all possible points of view; therefore certain findings may not be applicable to other settings. Another limitation relates to the shaping of current pregnancy experiences based on previous positive or negative pregnancy experiences and the degree of high-risk status of current and past pregnancies. This could not be adequately explored in our study, though was discussed throughout interviews as appropriate. Lastly, we did not collect data regarding whether patients were referred midway through their pregnancy care due to a specific high-risk reason by their family doctor, midwife or low-risk obstetrician, which also may have influenced perceptions.

### Future directions

This study validates the need to engage both HCPs and patients on their communication preferences in obesity-in-pregnancy clinical encounters. Future research may explore other settings beyond tertiary care, that may be less equipped for obesity care in pregnancy. Such a focus is important, since many of the negative experiences in healthcare experienced by participants occurred in healthcare centres accessed prior to being referred to a tertiary centre. This observation underscores the importance of developing effective communication tools that may be used across the full range of services accessed by pregnant individuals with obesity. At a later stage, this may also involve patients, psychologists, multidisciplinary clinical care providers and policy makers to identify solutions regarding communication between HCPs and patients as it pertains to weight. Another future direction includes integrating these challenges and solutions into medical education, which would be an integral point of improvement for the future of ethical and good healthcare delivery for these patients. Lastly, there is potential in this research area to explore the relationship between these narrative tensions with the different obesity categories and patient health characteristics (including BMI values). Differences in adiposity distribution and BMI categories may relate to certain patient experiences, and accordingly can be further explored.

## Conclusions

This layout of five major narrative tensions moves obesity-in-pregnancy research closer towards solutions to communication challenges, having a more precise tension in the conversations that take place in obesity-in-pregnancy clinical encounters. This paper does not provide a comprehensive list of recommendations, but highlights a series of tensions and how these might be amendable to practice, curriculum and policy change. Our in-depth exploration of patients' perspectives has important implications for improving patient care. Furthermore, the experiences of stigma described here offer specific points of reflection when it comes to challenging weight-related biases.

Although there may be competing narratives *between* the HCPs and the pregnant persons interviewed, it became apparent that narrative tensions also exist *within* the HCP participant group and *within* the patient participant group. This research introduces clarity on where, how and why HCPs and patients may be struggling in these clinical encounters, and adds knowledge on what to be aware of in terms of approaches to communication.

## Author Contributions

**Conceptualization:** Rachel Dadouch, Rory Windrim, Cynthia Maxwell, John Kingdom, Rohan D'Souza, Janet Parsons.

**Data curation:** Rachel Dadouch, Cynthia Maxwell, John Kingdom, Rohan D'Souza, Janet Parsons.

**Formal analysis:** Rachel Dadouch, Sarenna Lalani, Rory Windrim, Cynthia Maxwell, John Kingdom, Rohan D'Souza, Janet Parsons.

**Funding acquisition:** Rachel Dadouch, Rory Windrim, Cynthia Maxwell, John Kingdom, Rohan D'Souza, Janet Parsons.

**Investigation:** Rachel Dadouch, Rohan D'Souza, Janet Parsons.

**Methodology:** Rachel Dadouch, Sarenna Lalani, Rory Windrim, Cynthia Maxwell, John Kingdom, Rohan D'Souza, Janet Parsons.

**Project administration:** Cynthia Maxwell, John Kingdom, Rohan D'Souza, Janet Parsons.

**Resources:** Rory Windrim, Cynthia Maxwell, John Kingdom, Rohan D'Souza, Janet Parsons.

**Supervision:** Rory Windrim, Cynthia Maxwell, John Kingdom, Rohan D'Souza, Janet Parsons.

**Writing – original draft:** Rachel Dadouch, Rohan D'Souza, Janet Parsons.

**Writing – review & editing:** Rachel Dadouch, Sarenna Lalani, Rory Windrim, Cynthia Maxwell, John Kingdom, Rohan D'Souza, Janet Parsons.

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
