## [Decision Letter · Decision Letter 0]

29 Aug 2024

PONE-D-24-28024Where is communication breaking down? Narrative tensions in obesity-in-pregnancy clinical encountersPLOS ONE

Dear Dr. Dadouch,

Thank you for submitting your manuscript to PLOS ONE. After careful consideration, we feel that it has merit but does not fully meet PLOS ONE’s publication criteria as it currently stands. Therefore, we invite you to submit a revised version of the manuscript that addresses the points raised during the review process.

We look forward to receiving your revised manuscript.

Kind regards,

Rabie Adel El Arab

Academic Editor

PLOS ONE

Journal requirements: 1. When submitting your revision, we need you to address these additional requirements. Please ensure that your manuscript meets PLOS ONE's style requirements, including those for file naming. The PLOS ONE style templates can be found at https://journals.plos.org/plosone/s/file?id=wjVg/PLOSOne_formatting_sample_main_body.pdf and https://journals.plos.org/plosone/s/file?id=ba62/PLOSOne_formatting_sample_title_authors_affiliations.pdf. 2. We note that the grant information you provided in the ‘Funding Information’ and ‘Financial Disclosure’ sections do not match.  When you resubmit, please ensure that you provide the correct grant numbers for the awards you received for your study in the ‘Funding Information’ section. 3. Thank you for stating the following financial disclosure:  [Frederick Banting and Charles Best Canada Graduate Scholarships – Doctoral Research Award (CGS-D) to Honour Nelson Mandela Canadian Institutes of Health Research (CIHR), Canada Amount: $105,000 ($35,000 per year for 36 months)].  Please state what role the funders took in the study.  If the funders had no role, please state: ""The funders had no role in study design, data collection and analysis, decision to publish, or preparation of the manuscript."" If this statement is not correct you must amend it as needed. Please include this amended Role of Funder statement in your cover letter; we will change the online submission form on your behalf. 4. We note that your Data Availability Statement is currently as follows: [All relevant data are within the manuscript and its Supporting Information files.] Please confirm at this time whether or not your submission contains all raw data required to replicate the results of your study. Authors must share the “minimal data set” for their submission. PLOS defines the minimal data set to consist of the data required to replicate all study findings reported in the article, as well as related metadata and methods (https://journals.plos.org/plosone/s/data-availability#loc-minimal-data-set-definition). For example, authors should submit the following data: - The values behind the means, standard deviations and other measures reported;- The values used to build graphs;- The points extracted from images for analysis. Authors do not need to submit their entire data set if only a portion of the data was used in the reported study. If your submission does not contain these data, please either upload them as Supporting Information files or deposit them to a stable, public repository and provide us with the relevant URLs, DOIs, or accession numbers. For a list of recommended repositories, please see https://journals.plos.org/plosone/s/recommended-repositories. If there are ethical or legal restrictions on sharing a de-identified data set, please explain them in detail (e.g., data contain potentially sensitive information, data are owned by a third-party organization, etc.) and who has imposed them (e.g., an ethics committee). Please also provide contact information for a data access committee, ethics committee, or other institutional body to which data requests may be sent. If data are owned by a third party, please indicate how others may request data access.

Additional Editor Comments:

Dear Dr. Dadouch and colleagues,

Thank you for submitting your manuscript titled "Where is communication breaking down? Narrative tensions in obesity-in-pregnancy clinical encounters." Your research addresses an important issue in healthcare communication, particularly regarding obesity in pregnancy. While your study offers valuable insights, there are few areas that could benefit from further refinement. Below are my suggestions for improving the manuscript:

1. Methodology: Clarity and Rigor

Narrative Inquiry Approach: The use of narrative inquiry is appropriate for exploring the complexities of communication breakdowns in obesity-in-pregnancy encounters. However, the manuscript would benefit from more detailed descriptions of the data analysis process. Specifically, please provide a clearer explanation of how you identified and categorized the narrative tensions. This will enhance the transparency and rigor of your methodology.

Sampling and Reflexivity: While your sample size is reasonable for a qualitative study, it would be helpful to discuss any limitations this may present, such as the diversity of perspectives. Additionally, consider including a section on researcher reflexivity to acknowledge and address any potential biases that could have influenced the interpretation of the data.

2. Alignment of Objectives with Results and Discussion

Consistency: The objectives of your study—to identify where communication breaks down in obesity-in-pregnancy clinical encounters—are clearly aligned with the results. The identification of five narrative tensions provides a solid foundation for understanding these communication challenges.

Discussion Depth: While the discussion ties these narrative tensions back to existing literature, it would be beneficial to offer more specific, actionable recommendations for healthcare providers. Consider expanding the discussion to include practical strategies that can be implemented in clinical practice to address the identified communication challenges.

3. Enhance the Practical Implications

Actionable Recommendations: Your study highlights critical areas where communication breaks down, but the manuscript would benefit from more concrete suggestions on how healthcare providers can address these challenges. Providing detailed recommendations or strategies will help bridge the gap between theory and practice, making your findings more useful for clinicians.

Generalizability: Acknowledge the limitations of generalizability due to the specific context of your study (a single tertiary care hospital in Toronto). This will help temper the broader applicability of your findings and provide a clearer scope for their relevance in other settings.

4. Ethical and Power Dynamics Considerations

Power Imbalance: Your manuscript briefly touches on the power dynamics in patient-provider interactions but could benefit from a more in-depth exploration of these ethical considerations. Expanding on how healthcare providers can navigate these dynamics sensitively and ethically, especially when discussing weight-related issues, would add valuable depth to your discussion.

5. Use of Participant Quotes

Illustrative Quotes: Consider incorporating more participant quotes throughout the results section to better illustrate the narrative tensions and provide a richer understanding of the experiences being discussed. This will also help to balance the presentation of different perspectives within each narrative tension.

6. Overall Structure and Clarity

Reorganization for Clarity: Some sections of the results and discussion are quite dense. Breaking down complex paragraphs and ensuring each narrative tension is clearly introduced, discussed, and concluded will improve the readability and clarity of your manuscript.

Balanced Presentation: Ensure that all identified narrative tensions are presented with equal emphasis to provide a comprehensive view of the communication challenges in obesity-in-pregnancy encounters.

I look forward to seeing the revised version.

Best regards,

Reviewers' comments:

Reviewer's Responses to Questions

**Comments to the Author**

1. Is the manuscript technically sound, and do the data support the conclusions?

Reviewer #1: Yes

Reviewer #2: Yes

2. Has the statistical analysis been performed appropriately and rigorously? 

Reviewer #1: N/A

Reviewer #2: N/A

3. Have the authors made all data underlying the findings in their manuscript fully available?

Reviewer #1: Yes

Reviewer #2: Yes

4. Is the manuscript presented in an intelligible fashion and written in standard English?

Reviewer #1: Yes

Reviewer #2: Yes

5. Review Comments to the Author

Reviewer #1: The authors had conducted a qualitative study to understand the difficulties and the needs to have conversations with pregnant women on the impact of obesity during antenatal care and follow up.

Although the concepts and principles proposed are sound and appropriate, there are some points which I hope the authors can consider and revise in their manuscript:

1. It would be ideal to know more about the weight or BMI of these women who are going through these clinical encounters; this will likely to have "implications" on how these women are managed. There is likely to be clear differences in cases which are referred for direct obstetrician (i.e. specialist) care versus midwifery led care e.g. the very high BMI women may be referred for direct obstetrician follow up whereby there is already a selection bias or these women are prepped to understand and behave during each clinical encounter where possibly, they understand they are not under standard midwifery care if they are considered "high risk". The authors did not state this clearly and how this will impact the perceptions held during the consult and the frequency of follow up. Women with high BMI who are pregnant and have positive outcomes may also perceive these encounters differently or recorded by HCPs as low risk if their previous pregnancies are uncomplicated. This should be discussed and analyzed to allow this qualitative analyses to offer a more robust and balanced view.

2. There is no mention on how many of these women are under direct obstetrician care from the beginning or were then referred on for obstetrician care midway in their pregnancy because the nurses/midwife referred them in view of "higher" documented risks. This is important to explore as again, the impact on the perceptions on how the patients view and how the HCPs respond in accordance may drive the discussions.

3. There should be independent analyses of obstetricians consults versus nurses/midwives/others consults; there is a need to define who are the "others". This is because how the consults are conducted and the frequency of consults may affect on how the pregnant person responds in terms on the issues that HCPs are concerned about and hence the need for early discussions to ensure that the level of care and preparation are appropriate.

Reviewer #2: Interesting paper addressing a specific area of the management of obesity in pregnancy.

I would like to see included in the patient characteristics table the BMI data of the patients. Given that the patients were recruited from a specialized obesity-in-pregnancy clinic I suspect these patients are in the more severe end of the obesity spectrum. This info. would be give useful additional perspective to the reader.

6. PLOS authors have the option to publish the peer review history of their article (what does this mean?). If published, this will include your full peer review and any attached files.

Reviewer #1: No

Reviewer #2: No

---

## [Author Response · Author response to Decision Letter 0]

20 Oct 2024

Response to Reviewers

Dear editor and reviewers,

Thank you kindly for your time to review this manuscript and your feedback that we are looking forward to incorporating. Below you will see the replies to the points you have raised, initialled by Rachel Dadouch (RD) before the written response.

Thank you once again,

Rachel, on behalf of the authors of Where is communication breaking down? Narrative tensions in obesity-in-pregnancy clinical encounters

RD: I have updated the manuscript accordingly.

RD: We have corrected this accordingly in the platform. Note we have added more funding sources that were previously omitted. The revised information is as follows:

Funding Information:

Rachel Dadouch

Frederick Banting and Charles Best Canada Graduate Scholarships – Doctoral Research Award (CGS-D) to Honour Nelson Mandela Canadian Institutes of Health Research (CIHR), Canada Amount: $105,000 ($35,000 per year for 36 months) (2020-2023)

Ontario Graduate Scholarship Program (OGS) University of Toronto, ON (2019 – 2020)

Amount: $15,000 ($5,000 per semester)

Endowment Award Obstetrics and Gynaecology, University of Toronto Obstetrics & Gynaecology, ON (2019 - 2020)

Amount: $15,000

Bernard Ludwig Studentship in Obstetrics and Gynecology, Faculty of Medicine, University of Toronto, ON (2017 - 2019)

Amount: $20,000 (First Year) and $27,000 + top up $4,000 (Second Year)

Conference Research Travel Grant, School of Graduate Studies, University of Toronto, ON (2018)

Amount: $700

Financial Disclosure: 

Rachel Dadouch

Frederick Banting and Charles Best Canada Graduate Scholarships – Doctoral Research Award (CGS-D) to Honour Nelson Mandela Canadian Institutes of Health Research (CIHR), Canada Amount: $105,000 ($35,000 per year for 36 months) (2020-2023)

Ontario Graduate Scholarship Program (OGS) University of Toronto, ON (2019 – 2020)

Amount: $15,000 ($5,000 per semester)

Endowment Award Obstetrics and Gynaecology, University of Toronto Obstetrics & Gynaecology, ON (2019 - 2020)

Amount: $15,000

Bernard Ludwig Studentship in Obstetrics and Gynecology, Faculty of Medicine, University of Toronto, ON (2017 - 2019)

Amount: $20,000 (First Year) and $27,000 + top up $4,000 (Second Year)

Conference Research Travel Grant, School of Graduate Studies, University of Toronto, ON (2018)

Amount: $700

Rohan D’Souza’s research is supported through a Tier-2 Canada Research Chair in Maternal Health (CRC-2021-00337).

 [Frederick Banting and Charles Best Canada Graduate Scholarships – Doctoral Research Award (CGS-D) to Honour Nelson Mandela Canadian Institutes of Health Research (CIHR), Canada Amount: $105,000 ($35,000 per year for 36 months)]. 

RD: I have added the following to the cover letter: "The funders had no role in study design, data collection and analysis, decision to publish, or preparation of the manuscript."

RD: The datasets generated and analyzed (raw transcripts) during the current study are not publicly available due to ethical concerns and the importance of safeguarding the identities of our participants. Moreover, participants shared information on highly sensitive topics. Restricting access to the full interview transcripts is based on concerns regarding potential identifiability of the participants and the fact that they did not consent to share their data in a publicly available repository. Data are stored securely with the first author (RD). Researchers may contact the

corresponding author directly with any queries regarding data access.

It should be highlighted that these study findings cannot necessarily be replicated. Consistent with narrative inquiry, the data collection and analysis take on an interpretivist ontological approach, which allows for multiple interpretations. As per one of the reviewers’ comments, I have included more comments on reflexivity and the impact of these considerations on the study findings (lines 255-272).

RD: To our knowledge, this list is complete and correct.

Additional Editor Comments:

Dear Dr. Dadouch and colleagues,

Thank you for submitting your manuscript titled "Where is communication breaking down? Narrative tensions in obesity-in-pregnancy clinical encounters." Your research addresses an important issue in healthcare communication, particularly regarding obesity in pregnancy. While your study offers valuable insights, there are few areas that could benefit from further refinement. Below are my suggestions for improving the manuscript:

1. Methodology: Clarity and Rigor

Narrative Inquiry Approach: The use of narrative inquiry is appropriate for exploring the complexities of communication breakdowns in obesity-in-pregnancy encounters. However, the manuscript would benefit from more detailed descriptions of the data analysis process. Specifically, please provide a clearer explanation of how you identified and categorized the narrative tensions. This will enhance the transparency and rigor of your methodology.

RD: I have added details in the methodology data analysis section, found on page 9-10 (lines 217-245), to explain this process more thoroughly. It should be noted that the analysis of narrative inquiry is an interpretivist, non-linear process. It is conventional to provide a step-by-step overview of the analytic process that explains the transition from the development of themes and identifying patterns, to forming a broader story based on these patterns that we decided to assemble into narrative tensions. This is currently reflected in the manuscript.

Sampling and Reflexivity: While your sample size is reasonable for a qualitative study, it would be helpful to discuss any limitations this may present, such as the diversity of perspectives. 

RD: For this methodology, the diversity of perspectives is a welcomed facet of the work, that serves as a strength of the research. These narrative, in-depth interviews, are intended to be a deep focus into several individuals’ stories, rather than a wide collection of “everyone’s” stories. There is an accepted understanding that there will be plausibly more opinions, experiences and/or approaches to gather, had our interviewing continued, however, the strength and rigour of narrative work lies in diving deeply into the complexity of participants’ accounts. That said, we ensured we represented a wide range of clinicians and patients within our desired population (those involved in obesity-in-pregnancy clinical encounters), and that diversity, along with in-depth narrative findings, satisfy the methodological process and rigour. Because the sample size is not based on international representation nor does it reflect “every” point of view, the limitation is that this study is not generalizable to every patient in every context. Rather, we have provided a conceptual framework for understanding obesity-in-pregnancy clinical encounters in all their complexity and from multiple standpoints. We have edited and denoted that not all points of view are captured, in the ‘Strengths and limitations’ section (line 844). 

Additionally, consider including a section on researcher reflexivity to acknowledge and address any potential biases that could have influenced the interpretation of the data.

RD: We maintained reflexivity notes throughout this study. Without publishing too many personal details of the first author (RD – Rachel Dadouch) that may be relevant to reflexivity (e.g. ethnicity, gender, health status), I have added a few sentences to the Methodology section at the end of the data analysis subsection (lines 255-272).

2. Alignment of Objectives with Results and Discussion

Consistency: The objectives of your study—to identify where communication breaks down in obesity-in-pregnancy clinical encounters—are clearly aligned with the results. The identification of five narrative tensions provides a solid foundation for understanding these communication challenges.

Discussion Depth: While the discussion ties these narrative tensions back to existing literature, it would be beneficial to offer more specific, actionable recommendations for healthcare providers. Consider expanding the discussion to include practical strategies that can be implemented in clinical practice to address the identified communication challenges.

RD: We intentionally refrained from listing too many actionable recommendations (though some may be found in the implications subsection), because despite the rigour of this work, its purpose in part was to expose/illuminate the breakdown of communication and define the points of tension – as described in our edit on Line 822-826. Future research will be needed to explore specific communication strategies that patients would be comfortable with, which could differ in diverse cultural, geographical and contextual settings. Although this was beyond the scope of our study, we have included some suggestions for clinicians, such as on Line 746 where we explain the implications of patients viewing their pregnancies as high risk or not, and how clinicians should tailor their communication approach accordingly. 

3. Enhance the Practical Implications

Actionable Recommendations: Your study highlights critical areas where communication breaks down, but the manuscript would benefit from more concrete suggestions on how healthcare providers can address these challenges. Providing detailed recommendations or strategies will help bridge the gap between theory and practice, making your findings more useful for clinicians.

RD: Kindly see previous note on the limitations of our ability to provide recommendations based on this study alone.

Generalizability: Acknowledge the limitations of generalizability due to the specific context of your study (a single tertiary care hospital in Toronto). This will help temper the broader applicability of your findings and provide a clearer scope for their relevance in other settings.

RD: Thank you. Please see the end of the limitations section in the manuscript. In this work, the term ‘transferability’ is used instead of generalizability, in keeping with our study’s qualitative methodology.

4. Ethical and Power Dynamics Considerations

Power Imbalance: Your manuscript briefly touches on the power dynamics in patient-provider interactions but could benefit from a more in-depth exploration of these ethical considerations. Expanding on how healthcare providers can navigate these dynamics sensitively and ethically, especially when discussing weight-related issues, would add valuable depth to your discussion.

RD: Thank you for this suggestion – we elaborated on this more on line 816 to clearly elucidate the power imbalance that was mentioned earlier in the results.

5. Use of Participant Quotes

Illustrative Quotes: Consider incorporating more participant quotes throughout the results section to better illustrate the narrative tensions and provide a richer understanding of the experiences being discussed. This will also help to balance the presentation of different perspectives within each narrative tension.

RD: We included around 30 direct quotes and several more anecdotes in the results section, and in order to be succinct we were concerned that many more quotes would make for too long a manuscript. As mentioned in the following suggestion by the reviewer (that the results and discussion are at times overly dense), we wanted to avoid an overly dense publication. Although including more quotes can enrich the points made and bring to light the stories told, there are limitations for the purposes of the publication. Yet, based on this comment, we decided to add quotes on lines: 332 and 506. Thank you for this suggestion.

6. Overall Structure and Clarity

Reorganization for Clarity: Some sections of the results and discussion are quite dense. Breaking down complex paragraphs and ensuring each narrative tension is clearly introduced, discussed, and concluded will improve the readability and clarity of your manuscript.

RD: Thank you for this comment. We have edited accordingly with this piece of feedback in mind, such as through improving the formatting of narrative tension #1 and making narrative tension #5 more succinct.

[reviewer comment continued]

Balanced Presentation: Ensure that all identified narrative tensions are presented with equal emphasis to provide a comprehensive view of the communication challenges in obesity-in-pregnancy encounters.

RD: Originally, we attempted this, however the nature of the findings yielded an unbalanced emphasis on certain tensions over others. Accordingly, we wanted to represent the data in this way, in keeping with how partici

---

## [Decision Letter · Decision Letter 1]

23 Dec 2024

PONE-D-24-28024R1Where is communication breaking down? Narrative tensions in obesity-in-pregnancy clinical encountersPLOS ONE

Dear Dr. Dadouch,

Thank you for submitting your manuscript to PLOS ONE. After careful consideration, we feel that it has merit but does not fully meet PLOS ONE’s publication criteria as it currently stands. Therefore, we invite you to submit a revised version of the manuscript that addresses the points raised during the review process.

The manuscript has been re-evaluated by two reviewers, and their comments are available below. 

Could you please revise the manuscript to carefully address the concerns raised?

We look forward to receiving your revised manuscript.

Kind regards,

Helen Howard

Staff Editor

PLOS ONE

Journal Requirements:

Reviewers' comments:

Reviewer's Responses to Questions

**Comments to the Author**

1. If the authors have adequately addressed your comments raised in a previous round of review and you feel that this manuscript is now acceptable for publication, you may indicate that here to bypass the “Comments to the Author” section, enter your conflict of interest statement in the “Confidential to Editor” section, and submit your "Accept" recommendation.

Reviewer #1: All comments have been addressed

Reviewer #2: (No Response)

2. Is the manuscript technically sound, and do the data support the conclusions?

Reviewer #1: Yes

Reviewer #2: Yes

3. Has the statistical analysis been performed appropriately and rigorously? 

Reviewer #1: Yes

Reviewer #2: Yes

4. Have the authors made all data underlying the findings in their manuscript fully available?

Reviewer #1: Yes

Reviewer #2: Yes

5. Is the manuscript presented in an intelligible fashion and written in standard English?

Reviewer #1: Yes

Reviewer #2: Yes

6. Review Comments to the Author

Reviewer #1: Thank you for addressing my concerns and completed the necessary edits. The methodology is now clearer and it explains the authors' intention for the study and how it is conducted to achieve the outcomes observed.

Reviewer #2: Please include the BMI data in Table 1. This will give readers a better sense of the severity of the obese group recruited

7. PLOS authors have the option to publish the peer review history of their article (what does this mean?). If published, this will include your full peer review and any attached files.

Reviewer #1: No

Reviewer #2: No

---

## [Author Response · Author response to Decision Letter 1]

9 Jan 2025

Response to Reviewers

Dear editor and reviewers,

Thank you kindly for returning this manuscript, and for your feedback. I have outlined our responses and any changes below. The responses are initialled by Rachel Dadouch (RD) – see Pages 1 and 2. The clean and marked versions of the manuscript will be attached as well.

Thank you once again,

Rachel, on behalf of the authors of Where is communication breaking down? Narrative tensions in obesity-in-pregnancy clinical encounters

Journal Requirements:

Reviewers' comments:

Reviewer's Responses to Questions

RD: We have reviewed the reference list and made edits to entries (e.g. to links) that we noticed may be incorrect. We do not see any other incorrect reference entries beyond these changes. Kindly let us know if we have missed any errors. We did not remove citations throughout the last iteration; we arranged their order in our manuscript text.

Comments to the Author

1. If the authors have adequately addressed your comments raised in a previous round of review and you feel that this manuscript is now acceptable for publication, you may indicate that here to bypass the “Comments to the Author” section, enter your conflict of interest statement in the “Confidential to Editor” section, and submit your "Accept" recommendation.

Reviewer #1: All comments have been addressed

Reviewer #2: (No Response)

2. Is the manuscript technically sound, and do the data support the conclusions?

Reviewer #1: Yes

Reviewer #2: Yes

3. Has the statistical analysis been performed appropriately and rigorously?

Reviewer #1: Yes

Reviewer #2: Yes

4. Have the authors made all data underlying the findings in their manuscript fully available?

Reviewer #1: Yes

Reviewer #2: Yes

5. Is the manuscript presented in an intelligible fashion and written in standard English?

Reviewer #1: Yes

Reviewer #2: Yes

6. Review Comments to the Author

Reviewer #1: Thank you for addressing my concerns and completed the necessary edits. The methodology is now clearer and it explains the authors' intention for the study and how it is conducted to achieve the outcomes observed.

Reviewer #2: Please include the BMI data in Table 1. This will give readers a better sense of the severity of the obese group recruited

RD: Due to the nature of this study, we are hesitant to include the BMI data, as noted before on account of stigmatizing based on BMI values. This also relates to the limitations of BMI as a diagnostic tool for obesity, specifically with this study. For example, one may presume that a higher BMI is associated to some of these stories and findings, however the BMI value does not depict the nature of the obesity and the issues that may arise. Someone with a BMI of 31 may have abdominal adiposity that translates to the challenges during ultrasound appointments alluded to in this manuscript, whereas another patient with a BMI of 40 with their adiposity distributed elsewhere (and not abdominally) may not have these stories relevant to them. Including the BMI data for this narrative story can lead to readers falsely making conclusions about BMI categories. The stories and narrative tensions in this manuscript can apply to any individual with obesity, despite the BMI value. Kindly let us know if we can maintain our decision to omit the data on BMIs for the patient participants. To address this comment, we will however incorporate this consideration in our future directions. Please refer to Line 763, where we note that there is room in this research area to explore the relationship between BMI categories and the relevance of certain narrative tensions or stories: “Lastly, there is potential in this research area to explore the relationship between these narrative tensions with the different obesity categories and patient health characteristics (including BMI values). Differences in adiposity distribution and BMI categories may relate to certain patient experiences, and accordingly can be further explored.”

7. PLOS authors have the option to publish the peer review history of their article (what does this mean?). If published, this will include your full peer review and any attached files.

Do you want your identity to be public for this peer review? For information about this choice, including consent withdrawal, please see our Privacy Policy.

Reviewer #1: No

Reviewer #2: No

---

## [Editor Report · Decision Letter 2]

17 Jan 2025

Where is communication breaking down? Narrative tensions in obesity-in-pregnancy clinical encounters

PONE-D-24-28024R2

Dear Dr. Dadouch,

We’re pleased to inform you that your manuscript has been judged scientifically suitable for publication and will be formally accepted for publication once it meets all outstanding technical requirements.

Kind regards,

James Mockridge

Staff Editor

PLOS ONE

---

## [Editor Report · Acceptance letter]

30 Jan 2025

PONE-D-24-28024R2 

PLOS ONE

Dear Dr. Dadouch, 

I'm pleased to inform you that your manuscript has been deemed suitable for publication in PLOS ONE. Congratulations! Your manuscript is now being handed over to our production team.

Kind regards, 

on behalf of

Dr James Mockridge 

Staff Editor

PLOS ONE